# Endovascular Treatment of Cerebral Vein Thrombosis: Safety and Effectiveness in the Thrombectomy Era

**DOI:** 10.3390/diagnostics13132248

**Published:** 2023-07-03

**Authors:** Mariangela Piano, Andrea Romi, Amedeo Cervo, Antonella Gatti, Antonio Macera, Guglielmo Pero, Cristina Motto, Elio Clemente Agostoni, Emilio Lozupone

**Affiliations:** 1Department of Neuroradiology, Ospedale Niguarda Ca’ Granda, Piazza Ospedale Maggiore, 3, 20162 Milano, Italyguglielmo.pero@ospedaleniguarda.it (G.P.); 2Neuroradiology Unit, IRCCS Policlinico San Matteo, 27100 Pavia, Italy; 3Neurology and Stroke Unit, Department of Neuroscience, ASST Grande Ospedale Metropolitano Niguarda, Piazza Ospedale Maggiore, 3, 20162 Milano, Italyelioclemente.agostoni@ospedaleniguarda.it (E.C.A.); 4Department of Neuroradiology, Vito-Fazzi Hospital, 73100 Lecce, Italy

**Keywords:** cerebral vein thrombosis, mechanical thrombectomy, endovascular techniques, stroke

## Abstract

Cerebral venous thrombosis (CVT) is a rare cause of stroke that tends to affect young people. Endovascular treatment (EVT) has not yet shown to be beneficial in CVT and is therefore actually only indicated as rescue therapy in severe and refractory cases for medical treatment. Clinical, neuroimaging, procedural and follow-up data were evaluated in order to define the safety and efficacy of EVT in the management of CVT between January 2016 and December 2022. Safety was assessed on the basis of recording adverse events. Functional outcomes (NIHSS, mRS) and neuroimaging were recorded at onset, at discharge and at a 6-month follow-up. Efficacy was assessed evaluating the recanalization rate at the end of the procedure. Twenty-one patients (17 female, 4 male, range 16–84 years) with CVT underwent EVT. Overall morbidity and mortality were both at 4.7%. Median NIHSS at the onset and at the discharge were, respectively, 10 and 2. Successful recanalization was achieved in 21/23 procedures (91.3%). Imaging follow-up (FUP) showed stable recanalization in all but one patient with successful recanalization. In 18/21 patients, a good clinical independence (mRS 0–2) was recorded at 6 months. Our study adds evidence on the safety and efficacy of endovascular techniques in the treatment of CVT.

## 1. Introduction

Cerebral venous thrombosis (CVT) is a rare cause of stroke, accounting for 0.5–1% of all strokes [1,2]. It can occur in all ages, but it tends to present more frequently in the third decade and in female patients; estroprogestins, pregnancy and puerperium, anemia, obesity and pro-thrombotic disorders are known risk factors [3]. The spectrum of clinical presentation is wide, ranging from headache, nausea and vomiting to neurological deficit and coma depending on the extent and severity of the occlusion [4]. CVT carries a high risk of complications, namely hemorrhagic infarction and intracranial hypertension.

Systemic anticoagulation with heparin (unfractioned or with low molecular weight) is the standard medical therapy, even in patients with hemorrhagic lesions [5]. Endovascular techniques (thrombectomy or direct catheter thrombolysis) are reserved for drug-refractory cases [6,7,8]. In this scenario, many questions regarding mechanical recanalization need to be addressed, such as which subgroup of patients can benefit from the endovascular treatment or which is the correct timing for the endovascular treatment and how to evaluate technical EVT success most accurately.

Endovascular treatment of arterial ischemic stroke (AIS) had its breakout in 2015, when several controlled trials demonstrated significative improvement of the clinical outcome in patients undergoing mechanical revascularization compared to those receiving intravenous thrombolysis alone [9,10,11,12,13]. Conversely, only one randomized controlled trial was carried in order to determine the efficacy of the endovascular treatment of CVT [6]. The Thrombolisis or Anticoagulation for Cerebral Venous Thrombosis (TO-ACT) randomized controlled trial enrolled 67 patients from October 2011 to October 2016 to compare EVT in addition to best medical therapy for CVT. The trial was stopped early for futility, showing no significative differences in the outcome between the two groups.

Since then, technical knowledge of mechanical thrombectomy as well as device performance and availability have significantly grown over the years, improving the recanalization rate and the safety profile. Indications for arterial thrombectomy are constantly widening, including late-presenting or paucisymptomatic patents and distal occlusion [13,14,15,16].

On these grounds, we evaluated our case series of mechanical thrombectomies for CVT performed between January 2016 and December 2022.

## 2. Materials and Methods

### 2.1. Ethical Approval

All procedures were performed in accordance with the ethical standards of the institutional and/or national research committee and with the 1964 Helsinki Declaration and its later amendments or comparable ethical standards. Patient written informed consent was acquired before each procedure.

### 2.2. Study Design

This a retrospective, single-centre, observational study reporting the mechanical thrombectomies procedures for dural sinus thrombosis performed between January 2016 and December 2022. In this period, a total of 51 patients were referred to our Stroke Unit Department for CVT. Among these, 21 patients were classified as presenting with severe CVT and underwent endovascular therapy. Severe CVT was defined as the presence of one or more criteria between mental status disorder, neurological deficit, intracranial hemorrhage, deep venous system thrombosis or extensive involvement of dural sinuses according to the International Study on Cerebral Vein and Dural Sinus Thrombosis (ISCVT).

Inclusion criteria were:Severe spontaneous CVT;Neuroimaging of CVT;Availability of clinical and procedural data;Availability of clinical and neuroimaging follow-up.

Exclusion criteria were:Secondary causes of CVT as trauma or infection;Absence of clinical, procedural, neuroimaging or follow-up data.

Dural sinus thrombosis was diagnosed with a non-enhanced CT of the brain followed by a trifasic computed tomography angiography (CTA) scan in all patients.

Patient characteristics including risk factors, neurological signs and symptoms, procedural timing, technical and clinical outcomes were collected. Clinical, procedural and follow-up data were analyzed in order to evaluate the safety and effectiveness of the treatment.

Safety was analyzed recording intraprocedural and periprocedural (0–30 days after procedure) data in order to determine the morbidity and mortality rates. Morbidity was assessed with every severe adverse event (SAE, events with permanent clinical relevance) and mild adverse events (MAE, events with transient clinical relevance). Asymptomatic adverse events (AAE, events without any clinical relevance) and technical notes (TN) were also recorded. Clinical outcome was assessed recording NIHSS at onset and at discharge. 

Rescue thrombectomies were defined as procedures performed when, despite best medical therapy, patients showed clinical deterioration and/or thrombosis progression. Primary thrombectomies were defined as first-line procedures performed along with best medical management because of the clinical presentation and/or severe and extensive thrombosis.

Efficacy was assessed by evaluating the recanalization rate at the end of the procedure. Recanalization was classified into complete, partial and absent. Complete recanalization was defined as uninterrupted blood flow in the target sinus. Partial recanalization was defined as a narrowed or incomplete filling of the occluded sinus. Absent recanalization was defined as the lack of significant improvement of the flow within the sinus. Moreover, in the subcohort of patients with partial recanalization, an additional rating was assigned based on the presence or absence of cortical venous drainage delay. Successful recanalization was considered as complete or partial recanalization without a cortical venous drainage delay in the tributary territory.

### 2.3. Follow-Up

Clinical mid-term (6 months) and long-term follow-up was obtained for each patient with clinical examination or, when not possible, through telephone interview and assessed with a modified Rankin Scale (mRS).

Imaging follow-up was performed at discharge and at a 6-month follow-up with magnetic resonance venography and/or trifasic CTA in order to assess stability of the venous system patency.

### 2.4. Statistical Analysis

SPSS18.0 software was used to process the data. The enumeration data were tested by an χ2 test and expressed by n (%), and the measurement data were expressed by mean ± standard deviation (x ± sd). All the measurement data were in accordance with normal distribution. The mean values of the two groups were compared with two independent sample t-tests. Paired t-test was used to compare the mean before and after intervention in the same group, and repeated measured analysis of variance was used to analyze the repeated data. *p* < 0.05 indicated that the difference was statistically significant.

## 3. Results

### 3.1. Endovascular Technique

Mechanical thrombectomies were performed with local anesthesia and conscious sedation in 19/23 procedures; four procedures were performed under general anesthesia. An intravenous bolus of heparin (100 UI/kg) was administered in the angio-suite in patients who had not received anticoagulants or thrombolytics before the procedure.

A 5F short arterial sheath was placed in the right common femoral artery; a preliminary complete angiographic study was performed at the beginning of each procedure to evaluate the whole intracranial flow and venous drainage dynamics. A venous-phase image was obtained and used as a road map for the following steps of the procedure; arterial injections had a pivotal role, allowing the monitoring of recanalization of both occluded sinus and cerebral parenchyma transit time after each thrombectomy. Subsequently, an 8F right femoral vein access with a short sheath was performed, and a long-sheath 0.088″ catheter was navigated over a 180 cm 0.035″ guidewire and a 6F selective catheter to the internal jugular vein/sigmoid or transverse sinus depending on the site of occlusion. 

Both aspiration alone and a combined technique with a stentriever were adopted to obtain adequate recanalization of the occlude sinus. A large-bore aspiration catheter (0.071″–0.074″) was advanced up to the proximal extremity of the clot, connected to an aspiration pump and kept in that position for 1 min and then removed while simultaneously aspirating with a 60 mL vacuum lock syringe from the long sheath.

When a combined technique was performed, a 0.021″ or 0.027″ microcatheter was navigated inside the aspiration catheter on a 0.014″ guidewire to the distal part of the clot. A stentriever was delivered outside the microcatheter, opened in the occluded sinus and used as an anchor for the aspiration catheter to be moved up to the proximal end of the stent. The whole system was finally removed while aspirating from the long sheath. In one procedure, two stentrievers were opened in parallel and simultaneously removed. We used the largest diameter available for each stentriever given the large size of dural sinuses.

The number of attempts and materials chosen were at the discretion of a neurointerventionalist. Procedures were terminated when complete or partial recanalization of the sinus was obtained or when, despite multiple thrombectomies, there was no improvement in the venous drainage.

Median time from cerebral vein thrombosis diagnosis to groin puncture was 4 h (range 1–120 h); 15 patients were treated within the first day from the diagnosis, while 6 patients were treated 24 h or later or more from the diagnosis. 

Median procedural time was 123 min (range 40–340 min).

Aspiration alone was the first strategy in 20 out of 23 procedures; in 7 of these cases, thrombectomy was shifted to the combined technique with a stentriever, while the combined technique was adopted as first-line strategy in 3 out of 23 procedures.

### 3.2. Population

A total of 21 patients received endovascular treatment in the 6-year study period accounting for 23 procedures; patient characteristics are reported in Table 1. 

There were 17 females (81%) and 4 males (19%), median age was 44 years (range 16–84 years; median age for women was 44 years; median age for men was 45 years). 

Most common symptoms were headache (90.4%) and nausea or vomiting (57.1%); neurological deficits were present in 18 out of 21 patients (85.7%), with hemiparesis (47.6%) and aphasia (33.3%) being the most common. Seizures were observed in four patients (19%), seven patients (33.3%) had consciousness impairment.

In 14 cases (66.7%), two or more sinuses were involved; transverse sinus was the most common site of thrombosis (85.7%, Figure 1), followed by the superior sagittal sinus (71.4%). A total of 76% of patients had an intracranial hemorrhage (intraparenchymal hematoma, subarachnoid hemorrhage, or both).

The etiology/risk factors for CVT were identified in 17 patients. In nine patients, coagulation defects/disorders were found, while in six cases they were related to oral contraceptive drug assumption. One case occurred during the first trimester of pregnancy and in one case the CVT was timely related to neurosurgical intervention. 

The pregnant patient presented with a left hemisyndrome due to an extensive CVT involving the superior sagittal sinus, both the transverse and sigmoid sinuses. A single thrombectomy with a combined technique was performed, obtaining partial recanalization without venous delay; the patient recovered completely and delivered 6 months later without any complication or abnormalities in the newborn.

In 12/21 patients (57%), venous mechanical thrombectomy was adopted as a primary treatment along with best medical therapy.

### 3.3. Efficacy

Complete recanalization of the target sinus was achieved in 8 out of 23 procedures (34.8%), partial recanalization in 13 (56.5%), while in 2 cases there was no recanalization (8.7%). In all patients with partial recanalization (13/21), no residual venous drainage delay was observed at the end of the procedures.

Successful recanalization was achieved in 21 out of 23 procedures (91.3%) (Table 2).

### 3.4. Safety

Periprocedural complications occurred in two cases and consisted in a subdural hematoma due to sinus perforation; in the first case, it caused a significant increase in intracranial pressure in a patient with a large pre-existing intraparenchymal hemorrhage and a subarachnoid hemorrhage which required a decompressive craniotomy. In our series, this is the only subject who underwent more than one endovascular treatment because of a refractory CVT involving the SSS, both TS and the SS, who died 5 days after the first procedure because of the growth of intraparenchymal hematoma. The same patient required blood transfusion after the third procedure due to the large aspirated blood volume. In the second case, sinus perforation was complexly self-limiting (one asymptomatic adverse event, 4.7%) and subsequent subdural hemorrhage did not require any surgical operation due to the absence of mass effect on the cerebral parenchyma.

Overall morbidity and mortality were both at 4.7%.

### 3.5. Clinical Results

Median NIHSS at the onset and at the discharge (Table 3) were, respectively, 9.8 (range 0–24, CI 95% 6.7 to 12.9) and 3 (range 0–12, CI 95% 1.4 to 4.6). NIHSS at the discharge were significantly lower than the NIHSS at the onset (CI 95% −10.4 to −3.2; *p* < 0.0005). This trend was confirmed by comparison with pre- and post-procedural NIHSS of patients with complete recanalization (CI 95% −3.2 to −8.9; *p* < 0.005) and with partial recanalization (CI 95% −8.4 to −1, *p* < 0.05) of the CVT.

At the mid-term follow-up, a favorable performance status (mRS 0–2) was observed in 18 out of 20 surviving patients: 14 patients had an mRS of 0, 4 patients had an mRS of 1 and 2 patients had an mRS of 3. One of the two subjects with an mRS of 3 underwent mechanical thrombectomy without successful recanalization. The second one developed obstructive hydrocephalus secondary to the intraparenchymal hematoma that required external shunting. Medium time of last follow-up was 28 months.

No statistically significant differences were found between hemorrhagic and non-hemorrhagic patients, and between those receiving MT as a rescue or primary treatment.

### 3.6. Imaging Follow-Up

In all patients with successful recanalization (complete or partial), CTA or MR examinations performed at discharge and 6 months showed stable recanalization of the sinus in 19/20 among the surviving subjects. In one case (Figure 2), imaging follow-up showed re-occlusion of the right transverse sinus that was completely asymptomatic.

## 4. Discussion

In this retrospective study, we evaluate the safety and efficacy of mechanical thrombectomy for the treatment of CVT. Current guidelines [3,17] recommend full anticoagulation with LMWH or unfractioned heparin as standard treatment, followed by an oral vitamin K antagonist for 3–12 months to prevent recurrence [8,18]. Pharmacological therapy remains the gold standard for mild or moderate CVT. Endovascular treatment is currently considered as a second-line therapy in strictly selected cases. To date, many questions are still unaddressed regarding endovascular approach, such as which patients are eligible for mechanical thrombectomy, which is the right timing for EVT, and how to define technical success of EVT.

The TO-ACT trial [6] is the only randomized trial that compared EVT in addition to best medical management with best medical management alone in CVT subjects presenting at least one risk factor for poor outcome, but it was stopped early for futility.

No improvement in functional outcome at 12 months in patient outcome was demonstrated. Nevertheless, many limitations could have affected the results of the trial. Firstly, the patients were enrolled from 2011 to 2016 when physician expertise in clot retrieving techniques were limited. Moreover, the vast majority of the recent clot-retriever devices was not yet available; endovascular procedures were performed with AngioJet catheter plus local thrombolysis. Interestingly, the AngioJet device has been associated with lower recanalization rate and higher risk of complication [19].

Although AIS and CVT are fundamentally different diseases, the current situation of EVT for CVT in some ways resembles the setting of EVT for acute ischemic stroke a decade ago, when early EVT trials failed to show benefit. The knowledge of technical aspects of MT in CVT has significantly grown through the years, allowing neurointerventionalists to perform safer and more effective procedures, primarily deriving from the advancement in perfecting arterial thrombectomies [20,21].

In our experience, we reported a good safety and efficacy profile of the endovascular treatment of CVT. Successful recanalization was achieved in 91.3% of the procedures. Mortality and morbidity were both at 4.7%; these results are lower than those reported in the systematic review of Siddiqui et al. [19] (mortality 12%, morbidity 26%). Nevertheless, in the abovementioned review, patients had an overall more severe clinical presentation compared to our series, and a worse recanalization rate and higher complication rate was demonstrated in patients presenting with stupor or coma; this could suggest that endovascular recanalization may be more effective in the early stage of the disease. A significant limitation of the study is related to the inclusion of series acquired before the advent of mechanical thrombectomy in the AIS and thus the improvement of retrieving devices. Similar limitations affect the systematic review by Goyal et al., in which most of the studies were performed before 2015 [2], when clot retriever device availability was very limited and most of the procedures consisted in local thrombolysis and/or obsolete endovascular techniques such as clot maceration with guidewire or balloon thrombectomy. Another recent systematic review of the literature by Bucke et al. [22] included 21 studies, and only four of them started after 2015, when the most recent devices for arterial thrombectomy were available.

As previously mentioned, since the advent of endovascular treatment for AIS, there have only been few studies in the literature regarding thrombectomy for CVT.

Ma et al. [23] described 23 cases of CVT treated with stentriever alone, reporting a good safety and efficacy outcomes (no major complications and a 100% recanalization rate); however, functional performance status data were not reported. In the case series of Styczen et al. [24], 13 patients underwent MT for CVT with eligibility criteria and endovascular techniques similar to ours; recanalization rate (86%) and clinical outcomes (mRS 0–2 reported in 92% of the patients) were also comparable to our series.

Dandapat et al. [25] reported a good clinical outcome (mRS 0–3) at discharge in 9 out of 16 patients, although in this multicentric series MT was performed exclusively in refractory to medical therapy cases. A more recent, prospective and randomized study [26] compared the radiological and clinical outcome of patients with CVT undergoing mechanical thrombectomy with stentriever along with intrasinus thrombolysis and anticoagulant therapy with a control group that received thrombolysis and anticoagulants alone. The endovascular group showed significant improvement in neurological function at 7 days and a higher recanalization rate of the occluded sinus at 6 months; complication rate (e.g., ICH, headache, quadriplegia) was also lower in the interventional group.

In the same observational period of the present study, more than 1700 arterial thrombectomies were performed in our center; as the indications for ischemic due to large vessel occlusion keep on broadening, our eligibility criteria for endovascular treatment of cerebral vein thrombosis have been widening during the years and in relation to our increased confidence in clot retrieving (Figure 3). Indeed, the first patient of the present study had a more severe clinical setting at the time of the groin puncture compared to the most recent ones; this trend can be explained as the result of the growing knowledge and expertise achieved in arterial thrombectomies. Through the years in our center, endovascular treatment of CVT has been shifting from a rescue therapy to the primary approach in patients with risk factors.

The selection of patients that may benefit of the EVT for CVT remains a challenge that is still not resolved by the current literature [2,22]. The International Study on Cerebral Vein and Dural Sinus Thrombosis [4] identified male sex, age over 27 years old, mental status disorder, intracranial hemorrhage on admission, deep venous thrombosis, CNS infection and cancer as risk factors for poor outcome; in addition to these criteria, we considered thrombosis of two or more sinuses as a risk factor.

Our series differentiates from other studies [19,22,27,28,29] especially for door-to-groin timing; in 12 out of 21 patients, CT-to-groin time was less than 6 h. 

Endovascular recanalization of the sinus has an immediate impact on the cerebral venous drainage, reducing intracranial pressure and facilitating anticoagulants and/or thrombolytic effects on the clot; mechanical thrombectomy ensures a faster recanalization compared to medical therapy alone [4,5]. In patients with risk factors, this earlier recanalization can prevent thrombosis progression, reducing the risk of new intracranial hemorrhage, intracranial hypertension and thus morbidity and mortality [30]. On the other hand, performing mechanical thrombectomy exclusively as a rescue therapy—typically days after the diagnosis—in patients without symptom relief despite best medical therapy lowers the chance of an effective revascularization, since the clot structure changes and the clot becomes harder to remove for both endovascular and medical therapy [2,5,6]. In patients with pre-existing intraparenchymal hematoma, an early endovascular intervention can minimize fybrinolytics and anticoagulant dosage.

Moreover, a complete reperfusion may not be needed in CVT, but reduction in thrombus burden may be sufficient to allow autofybrinolysis to dissolve the residual amount of clot and achieve a favorable outcome [2]. Indeed, even a partially patent sinus can be sufficient to obtain a significant improvement on cerebral transit time, normalizing the arterio-venous pressure gradient. Parenchymal venous drainage times should be carefully evaluated after each thrombectomy with serial injections from the arterial catheter in order to avoid futile thrombectomies, thus reducing the complication rate. In the setting of acute CVT, even a small subdural hemorrhage can lead to a dramatic increase in intracranial pressure. In our series, further thrombectomies were performed in case of absence of poor improvements on venous drainage, or in cases with intraprocedural evidence of thrombosis recurrence or progression.

A large number of clots is usually removed after each thrombectomy, along with a significant volume of blood; blood loss and hemoglobin levels have to be constantly monitored, especially in patients undergoing more than one endovascular treatment.

Our study has several limitations since it is retrospective, monocentric and without a control group; furthermore, patient selection was heterogenous and evaluated on a case-by-case basis. 

## 5. Conclusions

To the best of our knowledge, this is one of the largest series of patients receiving exclusively mechanical thrombectomy as additional treatment to standard medical therapy for CVT.

Our series shows the safety and efficacy of both aspiration and combined technique with several devices for CVT; moreover, it shows the potential benefit of an early intervention in selected cases at high risk of morbidity and mortality. Along with the improvement in endovascular techniques, more studies and randomized controlled trials are required to better define patient selection and outcomes.

## Figures and Tables

**Figure 1 diagnostics-13-02248-f001:**
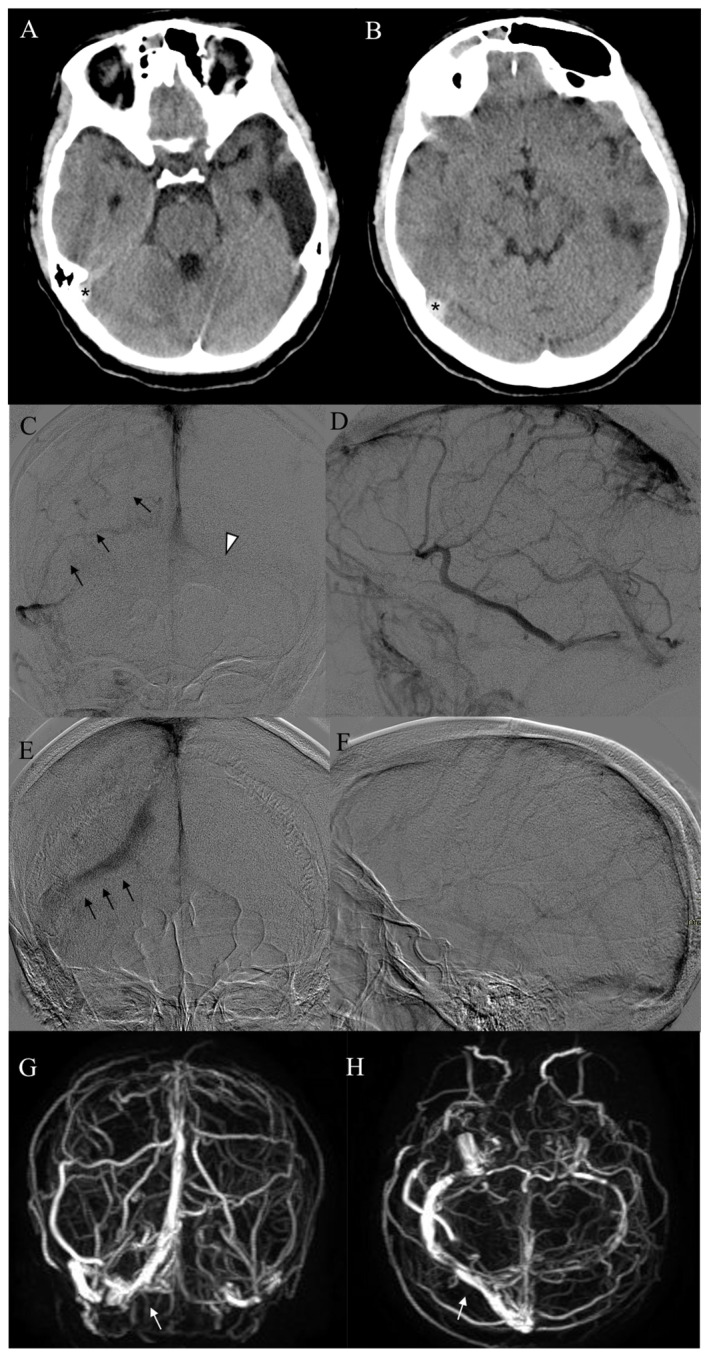
A 16-year-old female patient (Pt n. 17) with previous thrombosis of the left transverse sinus presents with headache and nausea. (**A**,**B**) Non-enhanced CT scan disclosed hyperdensity of the right transverse sinus (asterisks). (**C**,**D**) Pre-treatment frontal and lateral angiograms showed complete occlusion of the right transverse sinus (arrows) and hypoplasia of the contralateral transverse sinus (arrowhead). (**E**,**F**) Complete recanalization of the right transverse sinus (arrows) without cortical venous delay was achieved at final angiograms. (**G**,**H**) One-year MR follow-up demonstrated stable patency of the right transverse sinus (white arrows).

**Figure 2 diagnostics-13-02248-f002:**
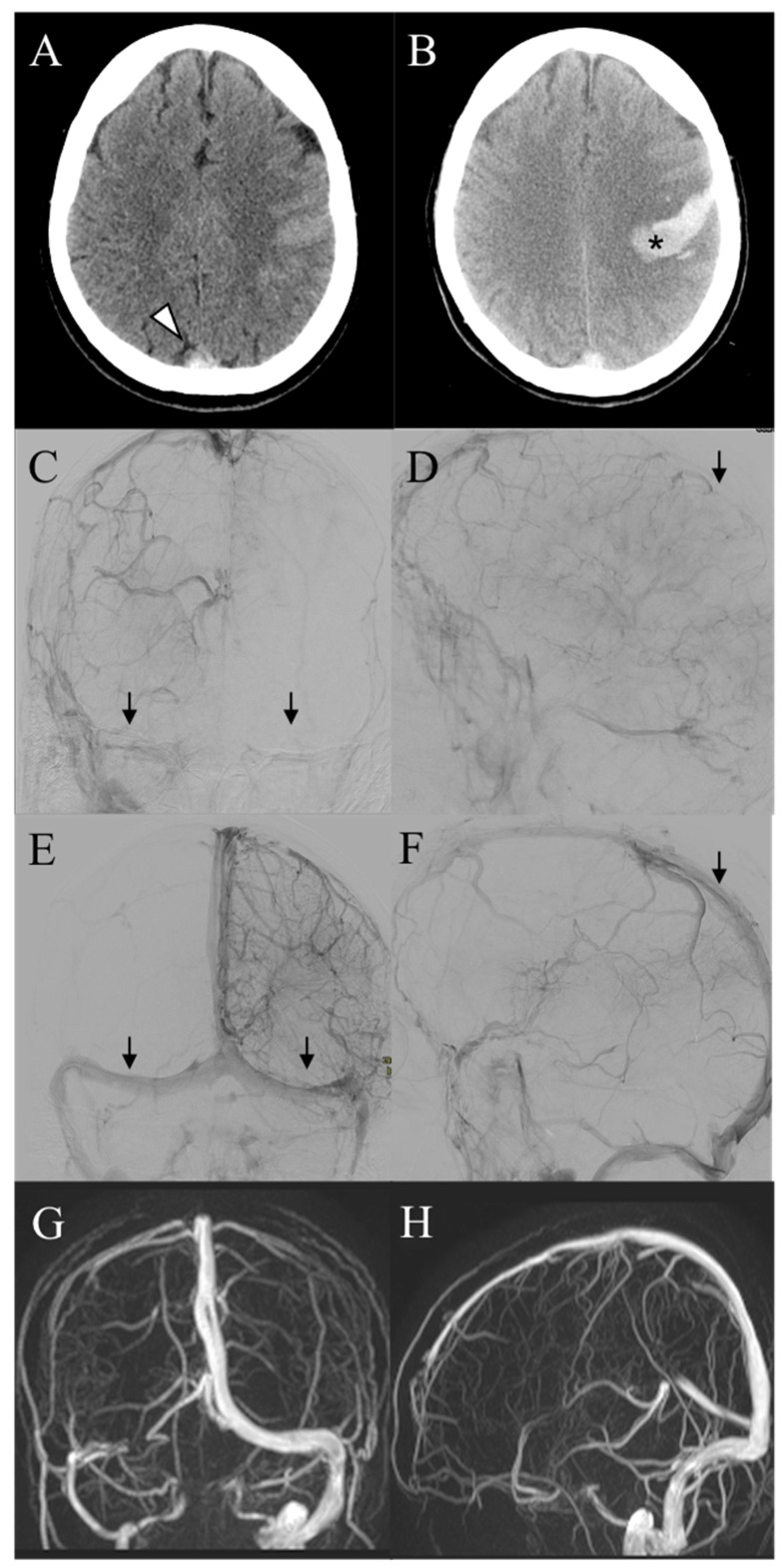
(**A**) A 40-year-old female patient (Pt n. 18) presents with headache and mild right hemisyndrome. CT scan showed spontaneous hyperdensity of the superior sagittal sinus (white arrowhead). (**B**) Progressive clinical worsening of the right hemisyndrome and consciousness impairment was recorded after 24 h despite medical therapy and the CT scan demonstrated hemorrhagic infarction in the left precentral region (asterisk). (**C**,**D**) Pre-procedural frontal and lateral angiograms demonstrated pan-thrombosis of the dural venous system (arrows) with a severe cortical venous drainage delay. (**E**,**F**) Final DSA after thrombectomy showed successful recanalization (arrows) and normalization of venous cortical drainage. (**G**,**H**) One year MR follow-up disclosed occlusion of the right transverse sinus, which was completely asymptomatic.

**Figure 3 diagnostics-13-02248-f003:**
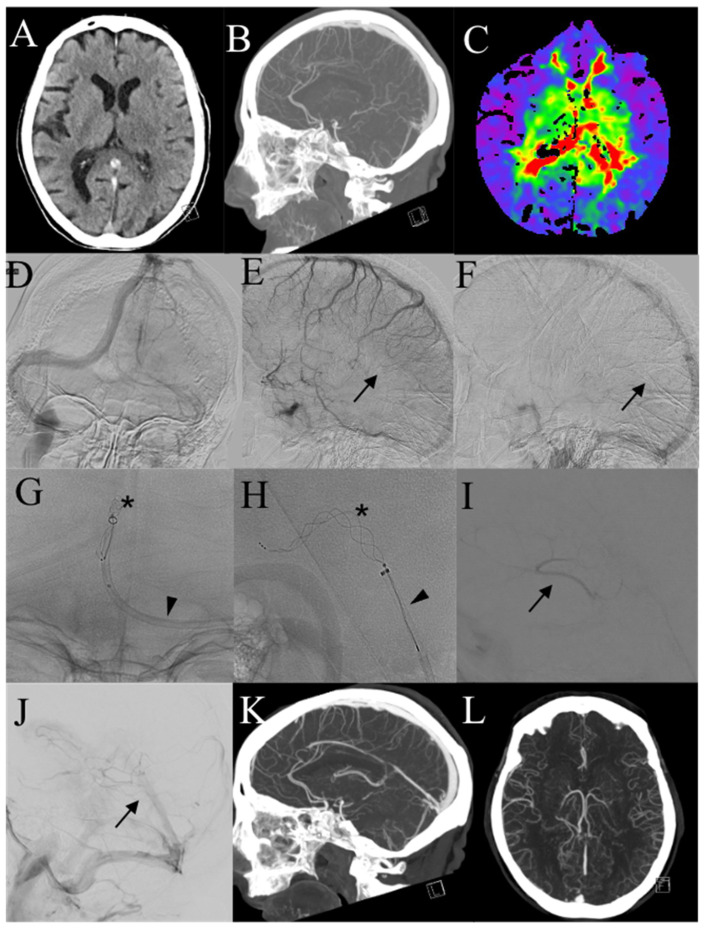
A 84-year-old patient (Pt n. 2). (**A,B**) Non-enhanced CT scan and CTA show hyperdensity of the straight sinus and thrombosis of the deep venous system. (**C**) Time-to-peak perfusion-CT map demonstrated significative contrast delay in the basal ganglia bilaterally. (**D**–**F**) Pre-treatment angiograms confirmed thrombosis of the deep venous system (arrows). (**G**–**H**) Combined technique with stentriever (asterisk) and aspiration (arrowhead) was performed in the straight sinus, achieving successful recanalization (**I,J**) of the internal cerebral veins and in the straight sinus. (**K,L**) One-year CTA scan demonstrated patency of the deep venous system.

**Table 1 diagnostics-13-02248-t001:** Patient characteristics.

	Sex, Age	Symptoms	Risk Factors	CVT Site	ICH	CT-to-Recanalization	Primary/Rescue	EV Techniques
1	F, 72	Headache, nausea, aphasia, rt hemisyndrome	No identifiable risk	SSS, SS, SrS, TS, VG	SAH	7 h	Rescue	Combined Aspiration plus Stentriever
2	F, 84	Headache, aphasia, rt hemisyndrome	No identifiable risk	TS, ICV, VG, TS, SS	No	5 h	Primary	Combined Aspiration plus Stentriever
3	M, 52	Headache, nausea, dizziness	Antiphospholipid antibodies +	TS	IPH, SAH	6 h	Primary	Aspiration
4	F, 51	Headache, rt hemisyndrome, seizures	Contraceptive pill, MTHFR mutation	SSS	IPH, SAH	2 h	Primary	Aspiration
5	F, 40	Headache, aphasia, seizures	Antithrombine III and Protein S deficiency, contraceptive pill	TS, SS, JV	IPH	7 h	Rescue	Aspiration
6	F, 44	Headache, lt hemisyndrome, seizures	No identifiable risk	SSS	SAH	1 h	Rescue	Aspiration
7	M, 41	Headache, nausea, photophobia, lt sup. arm deficit	ANA+	SSS	IPH	2 h	Primary	Combined Aspiration plus Stentriever
8	F, 44	Headache, aphasia	Contraceptive pill, activate protein C resistance	TS, SS	SAH	4 h	Primary	Aspiration
9	F, 76	Headache, seizures, loss of consciousness	No identifiable risk	SSS, SrS, TS (both), SS	No	5 h	Primary	Aspiration
10	F, 33	Headache, nausea, lt hemisyndrome	Pregnancy	SSS, SrS, TS (both), SS (both)	No	2 h	Primary	Combined Aspiration plus Stentriever
11	F, 49	Headache, vomiting, dizziness, aphasia, rt hemisyndrome	Recent pulmonary embolism	SSS, TS (both), SrS,	IPH, SAH	2 h	Primary	Combined Aspiration plus Stentriever
12	F, 49	Headache, nausea, dizziness, lt sup. arm deficit	Contraceptive pill, C677T and A1298C MTHFR mutation	Panthrombosis	No	2 h	Primary	Aspiration
13	F, 45	Headache	No identifiable risk	TS, SS	SAH	6 h	Primary	Aspiration
14	F, 44	Headache, nausea, aphasia, rt hemisyndrome	Contraceptive pill	TS	IPH	3 h	Primary	Combined Aspiration plus Stentriever
15	M, 83	Aphasia, rt hemisyndrome	Progestogen therapy	SSS, TS, SS, JV	IPH	24 h	Rescue	Combined Aspiration plus Stentriever
16	M, 49	GCS 9	Neurosurgery 9 days prior	SSS, TS	No	5 days	Rescue	Aspiration
17	F, 16	Headache, nausea	Previous contralateral CVT, Prothrombin gene mutation	SSS, TS, SS	No	2 h	Primary	Aspiration
18	F, 40	Headache, nausea, rt hemiparesis	Prothrombin gene mutation (A20210G homozygosis)	SSS, SrS, TS (both), SS	IPH	48 h	Rescue	Aspiration
19	F, 27	Headache, nausea, dizziness, lt hemisyndrome	Contraceptive pill	SSS, rt parietal vein	IPH	48 h	Rescue	Combined Aspiration plus Stentriever
20	F, 27	Headache, nausea, rt hemisyndrome	Thalassemia	SSS, lt temporal vein	IPH	48 h	Rescue	Combined Aspiration plus Stentriever
21	M, 19	Headache, nausea, vomiting, dizziness	Active protein C resistance, MTHFR mutation	TS, SS	IPH, SAH	3 h	Primary	Aspiration

ANA, Anti-Nucleus Antibodies; CVT, Cerebral Vein Thrombosis; ICH, Intra-Cranial Hemorrhage; ICV, Internal Cerebral Vein; h, hours; IPH, Intraparenchymal Hematoma; JV, Jugular Vein; Lt, Left; mRS, modified Rankin Scale; MTHFR, Methylenetetrahydrofolate Reductase; Rt, Right; SAH, Sub-Arachnoid Hemorrhage; SrS, Straight Sinus; SS, Sigmoid Sinus; SSS, Superior Sagittal Sinus; Sup, Superior; TS, Transverse Sinus; VG, Vein of Galen.

**Table 2 diagnostics-13-02248-t002:** Procedural outcomes (23 procedures performed on 21 patients).

	Efficacy	No. per Procedure
**Successful recanalization**		21/23 (91.3%)
	Complete recanalization	8/23 (34.8%)
	Partial recanalization without CVD * delay	13/23 (56.5%)
**Unsuccessful recanalization**		2/23 (8.7%)
	Partial recanalization with CVD * delay	0/23
	No recanalization	2/23 (8.7%)
	**Safety**	**No per Patient**
**Mortality rate**		1/21 (4.7%)
**Morbidity rate**		1/21 (4.7%)
	Severe Adverse Event (SAE)	1/21 (4.7%)
	Mild Adverse Event (SAE)	0/21
	Asymptomatic Adverse Event (AAE)	1/23 (4.3%)

* CVD: Cortical Venous Drainage.

**Table 3 diagnostics-13-02248-t003:** Patient outcome.

	Mean NIHSS at Onset ± SD	Mean NIHSS at Discharge ± SD	mRS 0–2	mRS > 2	mRS = 6
all patients (21)	9.8 ± 6.9	3 ± 3.5	18/21 (85.7%)	2/21 (9.5%)	1/21 (4.7%)
complete recanalization (8)	12.2 ± 6.1	3.3 ± 3.5	8/8 (100%)	0/8	0/8
partial recanalization (11)	6.4 ± 5.3	2.1 ± 2.3	10/11 (90.9%)	1/11 (9.1%)	0/11
no recanalization (2)	19 ± 5	27 ± 15	0/2	1/2 (50%)	1/2 (50%)

## Data Availability

The data presented in this study are available on request from the corresponding author. The data are not publicly available due to the privacy policy of our institution.

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
