# Peer review of "Endovascular Treatment of Cerebral Vein Thrombosis: Safety and Effectiveness in the Thrombectomy Era"

_diagnostics, 2023, doi:10.3390/diagnostics13132248_

Round 1
Reviewer 1 Report
The authors present an interesting single-center observational study about endovascular mechanical thrombectomy procedures for dural sinus thrombosis in 21 patients and 23 procedures. They demonstrate the safety and efficacy of this endovascular procedure with several devices for CVT and show the potential benefit of an early intervention in selected cases. The follow up period is very short (6 months) however considering the paucity of recent literature and reports about modern EVT techniques in CVTs , I would encourage to accept this paper.
Moderate editing of English language required
Reviewer 2 Report
Systemic anticoagulation with heparin is the main medical therapy for cerebral venous thrombosis (CVT). Whether the patients with CVT can benefit for endovascular techniques is still controversy. The manuscript shown the safety and efficacy of mechanical thrombectomy in a subgroup of patients with severe CVT, and the results indicated the potential benefit of an early intervention in the selected patients with severe CVT. The manuscript could be accepted after minor revisions.
1. The manuscript did not note the medical therapy for CVT patients.
2. In the Table1. 8/21 patients were noted “Rescue”, what’s the definition of “Rescue” or “Primary”, and when the endovascular techniques were used for “Rescue”?
Reviewer 3 Report
Thank you for this interesting read. This is a retrospective single-centre observational study on mechanical thrombectomies for CVST with 21 patients included in the analysis. Very clear study with a well-thought discussion. I only have a few minor remarks.
In the abstract it is unclear why in 21 patients there have been 23 procedures. Perhaps good to mention why this is the case. Please use the absolute numbers and percentages for the data (morbidity/mortality, mrs 0-2). Please correct the grammar in the sentence about morbidity and mortality. Please do not use FUP abbreviation without explanation, I suggest you write it out it when you first use it (also in the abstract).
Methods:
It is unclear to me what is you center policy on EVT in CVT. There were 51 patients with CVT and almost half underwent EVT even though this treatment is not recommended by any guideline and the only trial was stopped due to futility?
Also, if you assume all severe cases underwent CVT: severe CVT according to criteria can also be based on neurological deficit. Do all patients with any neurological deficit undergo EVT? Please explain in more detail how you select your patients?
Why do you choose to make a CTA and not a CTV/MR imaging?
Why do you choose to use NIHSS, do you think it is appropriate for measuring severity of CVT?
Who assessed whether a complication was MAE or SAE? How was it assessed whether a complication was or was not EVT related?
Results
Could you say something about the outcome of primary vs rescue thrombectomies? Are they different?
Also, are patients with intracranial lesions (hemorrhagic/non-hemorrhagic) benefiting more or less from the EVT? Are you able to say something about this?
Discussion
Very adequate discussion, raising crucial points. Needs re-reading and English corrections. Also, I think it is important to check what compare the outcomes to a historical cohorts of patients who did no undergo intervention. How do recanalization rates and outcomes compare? I think it is a crucial question to address - do we really need EVT if the outcomes without it are as good?
